# Current Status of Cell-Based Therapies for Vitiligo

**DOI:** 10.3390/ijms24043357

**Published:** 2023-02-08

**Authors:** Anna Domaszewska-Szostek, Agnieszka Polak, Monika Słupecka-Ziemilska, Marta Krzyżanowska, Monika Puzianowska-Kuźnicka

**Affiliations:** 1Department of Human Epigenetics, Mossakowski Medical Research Institute, PAS, 02-106 Warsaw, Poland; 2Faculty of Biology, University of Cambridge, Cambridge CD2 1TN, UK; 3Division of Ophthalmology and Optometry, Department of Ophthalmology, Collegium Medicum, Nicolaus Copernicus University in Toruń, 85-168 Bydgoszcz, Poland; 4Department of Geriatrics and Gerontology, Medical Centre of Postgraduate Education, 01-813 Warsaw, Poland

**Keywords:** vitiligo, keratinocytes, melanocytes, cell-based therapies, melanogenesis, cell transplantation

## Abstract

Vitiligo is a chronic pigmentary disease with complex etiology, the signs of which are caused by the destruction of melanocytes in the epidermis, leading to the lack of melanin pigment responsible for skin coloration. The treatment of vitiligo, which aims at repigmentation, depends both on the clinical characteristics of the disease as well as on molecular markers that may predict the response to treatment. The aim of this review is to provide an overview of the clinical evidence for vitiligo cell-based therapies taking into account the required procedures and equipment necessary to carry them out as well as their effectiveness in repigmentation, assessed using the percentage of repigmentation of the treated area. This review was conducted by assessing 55 primary clinical studies published in PubMed and ClinicalTrails.gov between 2000 and 2022. This review concludes that the extent of repigmentation, regardless of the treatment method, is highest in stable localized vitiligo patients. Moreover, therapies that combine more than one cell type, such as melanocytes and keratinocytes, or more than one method of treatment, such as the addition of NV-UVB to another treatment, increase the chances of >90% repigmentation. Lastly, this review concludes that various body parts respond differently to all treatments.

## 1. Background

Vitiligo is a chronic pigmentary disease that affects approximately 1% of the world’s population. This dermatological condition affects skin and hair and manifests itself by characteristic white macules and patches. The disease can be segmental, localized in one area of the body, or generalized, affecting a broader area. It can also be stable or unstable, depending on the appearance of new decoloration. Non-segmental vitiligo is prone to reactivation but, as in many of the studies, patients are required to have stable vitiligo, those with the segmental type of the disease are more likely to be selected for treatment. The loss of skin pigmentation is caused by the destruction of melanocytes in the epidermis, leading to the lack of melanin pigment responsible for skin coloration [1,2]. The published data suggest that the causes of melanin deficiency are multifactorial; mostly, it has an autoimmune origin with an underlying genetic tendency [3,4].

The disease has vast deleterious consequences beyond the aesthetical aspect, negatively affecting the patient’s emotional well-being and self-esteem. It is reported that around 75% of vitiligo patients have a psychological disorder [5], and female vitiligo patients experience significantly more significant impairment of general and psychological health, intimate relationships, and sexual function compared to healthy women [6].

The standard treatment for vitiligo varies based on the patient’s tolerance and consists of topical steroid therapy, immunosuppressant, sun protection, phototherapy, vitamin D, or narrow-band ultraviolet B (UVB) phototherapy (Figure 1). While these treatment options can be successful for certain patients, for others, they do not bring satisfactory results [7]. Moreover, some patients are resistant to conventional medical treatments [8], probably due to polymorphisms in the genes involved in the immune response and melanogenesis [9], which motivates the search for novel combination therapy.

Therefore, there is a great interest in introducing other novel therapies that are effective and safe for vitiligo patients (Figure 1).

This narrative literature review aims to bring a comprehensive knowledge of cell-based therapies in the treatment of vitiligo. In the present study, we discuss the development of these methods in recent years and point out various technical aspects affecting the effectiveness of the given methods for a better understanding of which procedure could be recommended, taking into account the characteristics of vitiligo and the patient’s age as well as facilities available in the clinic. Different transplantation approaches using keratinocytes and melanocytes or both cell types are presented with a description of the most applicable techniques such as hair follicle cell transplantation, the ReCell system, the Jodhpur Technique, and an approach where cell transplantations are combined with narrowband ultraviolet B (NB-UVB) or autologous platelet-rich plasma.

This literature review aims to bring knowledge of cell-based therapies by assessing 55 of the most effective primary clinical studies published in PubMed (https://pubmed.ncbi.nlm.nih.gov/, accessed on 20 November 2022) [10] and ClinicalTrails.gov (https://clinicaltrials.gov, accessed on 20 November 2022) [11] between 2000 and 2022 with the oldest studies being discussed for historical relevance. Moreover, this review aims to give an indication of the therapies depending on the vitiligo type and location. Only human-based clinical studies and case reports were included in this review.

This review is divided into five sections, each considering a different cell-based therapy, its background, studies, and effects. It should be stated that the studies presented here have been conducted on several different ethnic groups. It is known that people of varying skin colors respond differently to vitiligo treatment; thus, this must be kept in mind when analyzing the results of clinical trials.

## 2. Melanocytes and Keratinocytes as the Targets for Vitiligo Therapy

Neural crest cell-derived melanocytes are the melanin-producing cells of the skin; several melanocyte cell death mechanisms have been proposed to explain the origin of vitiligo. As such, the transplantation of healthy cells shows great promise for treating vitiligo patients. Several methods for the delivery of non-cultured melanocytes into the affected skin areas of patients have been attempted [12,13,14], including transplantation onto dermabraded or laser-abraded areas. In this approach, the skin sample is shortly incubated with trypsin and centrifuged before spreading on the recipient area. As the number of melanocytes in this method is not increased in culture, its efficacy might be lower compared to cultured melanocyte transplantation. This is suggested by the relatively low (57.4%) number of patients achieving >50% repigmentation in the study by Ghorbani et al. [14]. Melanocytes only account for up to 10% of skin cells and thus should be amplified in vitro prior to transplantation to maximize the chances of success [15].

Therefore, there are numerous approaches for transplanting pure cultured melanocytes [13,16,17,18]. For example, in the Chen et al. study [16], 25 segmental vitiligo patients were treated with cultured autologous melanocytes. The cells were transplanted into laser-denuded areas at a density of 70,000 to 100,000 melanocytes per cm^2^. Complete repigmentation was seen in most patients in less than one month, although in some cases, a thin vitiliginous line at the junction between normal skin and the transplant was visible. Hair follicles, the outer root sheath, are rich in melanocytes with potential proliferative ability. Thus, this offers a potential donor site for autologous cell transplants and was recently explored in several studies [19,20,21,22,23]. In the study by Shi et al. [23], the occipital area was used to obtain a scalp specimen containing at least 15 hair follicles. Following the removal of adipose tissue, the remaining hair follicles and dermal papillae were incubated, and a single-cell suspension was created. The sterilized recipient area was then abraded superficially using a motorized dermabrader or a CO_2_ laser and covered with the cell suspension and a hyaluronic acid dressing. In the nine months following treatment, 22 of the 26 patients achieved >75% repigmentation, and of those, 9 individuals achieved >90% repigmentation. Moreover, hair follicle transplantation using the hair follicular unit transplantation (FUT) technique is a cheap, simple method requiring minimal infrastructure, which makes it suitable for small stable lesions affecting hairy body parts [24].

Melanocytes can also be transplanted using dermarolling treatment, which involves microneedles piercing the epidermis for cell delivery. Melanocytes can be obtained with superficial shaving of scalp skin, as this is another area known to be particularly rich in these cells. Following incubation in a trypsin solution, the epidermis can easily be separated from the dermis using forceps. The epidermis is then centrifuged, the supernatant removed, and the pellet suspended in plasma. Following cleaning with an antiseptic spray, the recipient site can be dermarolled to deliver the melanocytes. Benzekri and Gauthier [25] have shown that after 24 h, nearly all holes had closed up without signs of infection, and melanocytes had been observed in the basal layer of the epidermis. After six months, 40% of the patients had an excellent response (76–100%) to the treatment. Autologous melanocytes can also be obtained from the thigh or buttock areas with normal skin color, which was explored in generalized vitiligo patients. The autologous material was incubated with trypsin then mixed with patients’ serum and centrifuged. The cell suspension mixed with hyaluronic acid was then evenly spread on recipient areas, previously injected with lidocaine, and shaved with a curate device. The success of the treatment was highly dependent on the recipient area, with the highest proportion of excellent and good results achieved in various areas of the face (57.4% on the face compared to the overall success of 50%) [14]. Using eyelid skin to harvest melanocytes and subsequent transplantation of the autologous melanocytes yielded similar results, with >80% repigmentation in 56% of the cases. The high success rate could be in part due to the selective growth of melanocytes while inhibiting fibroblasts and keratinocyte cells, as their faster growth and high proportion prevent the growth of melanocytes. The best results and even coloration were achieved in the legs, trunk, and face, and it was observed that sunlight could promote pigment production of transplanted cells [26]. Even though the results of the three latter studies are satisfactory, using hair root melanocytes appears to be the most effective solution. Moreover, it is clear that the choice of melanocyte donor area for melanocytes should be influenced by the recipient area affected by vitiligo, as the different methods showed varying success in various locations on the body [27]. Interestingly, Zhu et al. [26] found a higher level of anti-melanocyte antibodies in the vitiligo patients’ serum, indicating that humoral immunological mechanisms could play a role in the development of the disease.

The clinical characteristic of vitiligo is another important aspect determining the effectiveness of transplantation. One hundred and twenty cases of vitiligo patients were studied, and it was investigated whether stable localized vitiligo, stable generalized vitiligo, and active generalized vitiligo show different outcomes after cultured autologous pure melanocytes transplantation. In this study, similar to previous work of this group [16], 60,000 to 100,000 melanocytes/cm^2^ were applied on the skin after carbon dioxide laser abrasion of the vitiligous areas. The best outcome was observed in the stable localized vitiligo group, where 84% of patients achieved 90% to 100% repigmentation. An excellent percentage of coverage was shown in 54% of patients in the stable generalized vitiligo group and none in patients suffering from active generalized vitiligo. This study proved the validity of the treatment of stable vitiligo with cultured autologous pure melanocytes [28]. The clinical applications of melanocyte cell transplantation in vitiligo are summarized in Table 1 and Figure 2. However, most clinical trials attempt to transplant keratinocytes in co-culture with melanocytes [29,30].

## 3. Melanocyte–Keratinocyte Cell Transplantation (MKCT)

It should be clarified that MKCT is the complete clinical grafting procedure which includes harvesting epithelium from the donor site, preparing the recipient site, and applying the suspension and dressing the wound, whereas non-cultured epidermal suspension (NCES) refers to a prepared cell suspension used in MKCT. The first introduction of non-cultured epidermal cellular grafting in the treatment of stable vitiligo took place in 1992 [31] after several successful attempts under experimental conditions on piebald guinea pig skin [32]. In this treatment, both melanocytes and keratinocytes are transferred, as melanocytes grow better in the presence of keratinocytes and produce better repigmentation. For instance, Phillips et al. [33] demonstrated the significance of improving the method of maintaining melanocyte numbers by introducing a feeder layer. The use of a hyaluronic acid-enriched cellular graft gave a repigmentation rate of over 70% in the vitiligous areas in 77% of patients after 12 months compared to a placebo in a double-blind study [1]. For a change, Khodadadi et al. [34] replenished the missing melanocytes and keratinocytes using a different route of their administration: the cell suspension was injected intraepidermally into vitiligous lesions. In the 6-month follow-up, 4 out of 10 patients had achieved moderate repigmentation (76–100%) and one patient’s patch was fully repigmented. The authors found no correlation between the number of transplanted cells and the outcome. Further development of this technique gave a repigmentation rate of over 50% in 32.2% of treated patches, whereas acquired repigmentation remained stable in 79.3% of treated patches during the 30-month-long follow-up period. Observing 300 patients, the first pigmentation loss in treated patches started around 9 months post-transplantation and mostly occurred during the first year (68.5%, n = 150) after treatment [35]. It is worth mentioning the results of the study by Budania et al. [34] and Bao et al. [36] which compared the NCES method with suction blister epidermal grafting (SPEG) and showed a better extent of repigmentation after NCES. Interestingly, in the study by Budania et al. [37], no melanocyte culture media, trypsin inhibitor, or hyaluronic acid was used, and only simple syringe-base suction was applied. Moreover, a comparison between an autologous non-cultured extracted hair follicle outer root sheath cell suspension (NCORSHFS) and NCES showed comparable efficacy in repigmentation [38,39], although, patients in the NCES group were significantly more satisfied than the patients in the NCORSHFS group [38]. However, there is also a study that suggests that cultured melanocyte transplantation (CMT) may give better repigmentation as compared with NCES in the case of stable generalized and segmental vitiligo [40]. Interestingly, a superior repigmentation to NCES or NCORSHFS alone was achieved when those two techniques were combined. The authors suggested that this approach may be a good alternative for the more resistant-to-treatment acral vitiligo [41].

The MKCT grafting procedure was also substantially developed since its discovery by Olsson and Juhlin in 1998 [42]. Initially, the sample of superficial skin was removed, and cells were isolated, separated, and cultured in a melanocyte growth medium. To carry out this procedure in one day, the next step was to apply the melanocyte-enriched epidermal cell suspension directly on dermabraded depigmented skin. Some of the changes proposed by Mulekar [43] concern the use of Dulbecco’s Modified Eagle’s Medium (DMEM) and Ham’s F-12 Nutrient Mixture for cell separation procedure and CO_2_ incubator substitution with an ordinary incubator. The CO_2_ incubator helps to maintain the pH in the cell cultures; however, it makes the procedure more expensive. Currently, melanocyte–keratinocyte cell transplantation (MKCT) involves obtaining a skin biopsy from the donor site one-tenth of the recipient area size. This is followed by soaking it in trypsin-EDTA solution, separating the dermis from the epidermis, and disposing of the dermis. The sample is then centrifuged, and the stratum corneum is discarded. Finally, the cell suspension is transplanted onto the deep epidermis of the dermabraded recipient area, which is then covered with a dry collagen sheet. The whole treatment can be conducted as a 2 to 4 h outpatient procedure [44].

In this technique, repigmentation can be seen between 2 weeks and 2 months after surgery [45]. Many patients show hyperpigmentation, but it usually blends with the surrounding skin in 6 to 8 months, and the likelihood increases when patients expose the transplanted areas to sunlight [43]. This method is most effective in segmental and focal vitiligo patients, for whom a scarring or cobblestone appearance is unlikely. Interestingly, the only post-operative pain that can be observed is in the feet and ankles [43]. Six months following the surgery, 84% of the patients showed good to excellent repigmentation, and in the long term, six years after the surgery, the treatment results remained positive for patients with segmental, stable vitiligo with the absence of fingertip involvement [46]. Mulekar et al. [47] also evaluated the effects of this treatment in 49 patients with segmental vitiligo and 15 patients with focal vitiligo who were followed up for up to 5 years. Overall, 95% to 100% of repigmentation was achieved in 84% of patients with segmental vitiligo and 73% of those with focal vitiligo, while a poor outcome was observed in 10% and 20%, respectively. Another study using MKCT reported that in the 12 to 72 months post-treatment, good to excellent repigmentation remained in 71% and 54% of patients with stable and non-stable vitiligo, respectively, confirming the success of this treatment for stable vitiligo patients. Interestingly, at 12 months, 62% of patients showed additional repigmentation that was not present before, and only 26% showed partial or complete regression. It was noted that improvement peaked at 10 months post-surgery and stabilized by up to a year, plateauing at around 18–24 months [44]. In another paper by Mulekar et al. [48], 142 patients were followed up to 6 years after autologous, non-cultured melanocyte-keratinocyte cell transplantation. Complete repigmentation was shown in 56% of patients, while poor pigmentation was observed in 24%. Another group [49] presented data concerning three cases of patients with stable genital vitiligo. A 26-year-old male with the loss of pigmentation on the penis glans and neck, and 24- and 39-year-old males with depigmentation of the glans and shaft of the penis, all of whom were treated with autologous, non-cultured MKCT. All of the patients achieved almost complete repigmentation. In another study [50], patients with stable vitiligo were treated using the same method as Mulekar and co-workers [43]. The outcome of the treatment of eight vitiligous patches treated with autologous non-cultured melanocyte–keratinocyte transplantation was compared to six control lesions, which were only dermabraded. The results were evaluated after 4 months. Over 95% repigmentation was observed in 50% and 0% to 24% repigmentation in 37% of patients treated with MKT. Five out of six control patches failed to show repigmentation, and one patch resulted in hyperpigmentation following inflammation.

Vazques-Martinez et al. [51] and Quezada et al. [52] compared the efficacy of the transplantation of melanocyte and keratinocyte (MKCT) cell suspension after dermabrasion (DA) or with dermabrasion only. In the 12-month follow-up period in Vazquez-Martinez’s study [51], there was no statistically significant difference between MKCT + DA and DA alone in the area of depigmentation but clinically MKCT + DA showed slightly better results. Quezada et al. [52] analyzed the results 3 months after transplantation and observed no significant differences between the treatments. It should be noted that the two latter studies indicate that MKCT + DA and DA alone are similar in terms of efficacy, whereas Mulekar and co-workers have found MKCT to be significantly better. This highlights the differences between methods and samples used in different studies even when, in principle, the technique used is the same.

Huggins et al. [53] performed 29 MKCT procedures in 23 patients with focal, generalized, or segmental vitiligo with no control group. Overall, 52% of the vitiligous lesions were localized on the extremities. Patients were evaluated monthly between the 3rd and 6th month after the procedure. Excellent repigmentation was achieved in 17% and poor in 41% of patients. The presence of vitiligo on the face/neck was associated with a better response to the treatment, with 19% of patients showing excellent repigmentation and 50% good.

Ebadi et al. [54] conducted a study where patients with stable generalized vitiligo, with a total of 39 patches, were divided into four groups. Nine patches were treated with MKCT alone, ten patches with MKCT and excimer laser, ten patches with excimer laser alone, and ten patches served as controls and did not receive any treatment. The authors reported that excimer laser combined with non-cultured MKCT improves the repigmentation rate, with an average of 41.9% reduction in the depigmented area surface. For comparison, it was only 4.7% for patients treated with excimer laser alone, 15.9% for those treated with MKCT alone, and 0.1% for the control group.

Lastly, it has been observed that the ethnicity of patients must be considered in the choice of MKCT treatment, for instance, Asian patients are more prone to hypertrophic scarring [30]. Even though vitiligo affects all ethnic groups similarly, it may be more noticeable in people with darker skin, and some treatments may show varying efficacy between the different groups.

It is worth mentioning that the response to NCES treatment may depend on the location of the lesions. The best results were observed in face and neck lesions (88% of satisfactory responses) and the worst in the extremity lesions (33% of satisfactory responses). However, based on their results, Mulekar et al. [55] concluded that the concept of a “difficult-to-treat site” is a relative term and depends upon the technique used. The non-cultured MKCT seems to be favorable as this technique does not require special precautions to treat these anatomical sites. According to some authors, using a higher density of melanocytes in the suspension [56], a strict immobilization procedure for the treated areas, and post-operative phototherapy in the form of sun exposure might improve the results in such a “difficult-to-treat site” [57]. Others reported that a better response was achieved in segmental vitiligo patients compared to non-segmental ones (84% compared to 63%) [58]. Moreover, NCES was also successfully used in the repigmentation of leucotrichia in vitiliginous patches. The proposed mechanism of this process is a retrograde migration of transplanted melanocytes or interfollicular epidermal stem cells to the hair bulb and/or their production of cytokines, which stimulates melanogenesis in the follicular bulbs [59]. It should also be mentioned that skin grafting may be considered to treat localized vitiligo in children [60,61]. In this group of patients, an MKTP is favorable as it does not require a long surgical time, and there is no need for absolute immobility and any special precautions to treat ‘‘difficult-to-treat’’ sites. The clinical applications of MKCT in vitiligo are summarized in Table 1 and Figure 2.

## 4. ReCell System in Vitiligo Therapy

ReCell is a robust point-of-care autologous therapy designed to treat skin defects such as small and large thermal burn wounds using a patient’s regenerative cells. The ReCell system enables the harvesting of autologous cells, processing them, and delivering them using a spray applicator. Three clinical trials analyzed the results of treating patients with stable vitiligo with ReCell (a cell suspension with keratinocytes, melanocytes, dermal papillary fibroblasts, and Langerhans cells sprayed over the wound) [62]. Mulekar et al. [63] compared the efficacy of the ReCell system and melanocyte–keratinocyte transplantation 4 months after the procedure. In both methods, the cell suspension was spread to previously dermabraded areas, and the results of the treatments were comparable.

Due to efficacy, time, and cost, surfaces that can be covered with cultured melanocytes are larger than those that can be covered with non-cultured cells. Cervelli et al. [62] treated 15 patients, and 12 of them (80%) achieved more than 75% repigmentation. The authors observed an excellent color match in 66% of patients.

In 2010, the same group [64] presented a case report of a 30-year-old man suffering from stable vitiligo on his hands. Before undergoing ReCell therapy, he had vitamin A, C, E, and UVB therapy, none of which were beneficial. Treatment with the ReCell system gave excellent results, both in the extent of repigmentation and the skin color match. The clinical applications of the ReCell system in vitiligo are summarized in Table 1 and Figure 2.

## 5. Autologous Non-Cultured, Non-Trypsinized Epidermal Cell Grafting

This method is also known as the Jodhpur Technique (JT) (the first time was carried out in the Medical College in Jodhpur in India) and is a modification of the autologous non-cultured, non-trypsinized keratinocyte–melanocyte cellular graft technique. The grafting material is rich in melanocytes and is obtained following the dermabrasion of the donor area. The epidermal particles fragmented during dermabrasion become entangled in an ointment, and a paste-like material is obtained. This material is laid out over the recipient lesion area using a graft spreader. This technique is very low-cost and does not need any sophisticated equipment.

The study by Tyagi et al. [65] with the use of the JT technique revealed that in both epidermal cell suspension and epidermal curettes, over 75% repigmentation was achieved in 60% of lesions. Moreover, the color matching with surrounding skin and yield of grafts was not significantly different between these techniques. Even better results were presented in the study by Lamoria et al. [24], where excellent repigmentation (>75%) was observed in 70% of lesions. In terms of the repigmentation rate, side effects, patient satisfaction, and dermatology life quality index reduction, this method was superior to FUT. The clinical applications of non-cultured epidermal cell grafting in vitiligo are summarized in Table 1 and Figure 2.

## 6. Cell Transplantations in Combination Therapy with a Narrowband Ultraviolet B (NB-UVB) or Autologous Platelet-Rich Plasma

Narrowband ultraviolet B (NB-UVB) is one of the treatment options for patients suffering from active vitiligo [66]. It promotes the proliferation and migration of cultured melanocytes. Zhang et al. [67] collected a group of 473 patients and investigated the effect of NB-UVB in combination with autologous melanocyte transplantation. The patients were divided into four groups: group 1 underwent NB-UVB sessions before melanocyte transplantation, group 2 was NB-UVB treated after transplantation, group 3 received NB-UVB before and after transplantation, while group 4 did not undergo NB-UVB sessions and received only transplantation. The best results were observed in group 3, where ≥90% repigmentation was achieved in 81% of patients, which suggests that NB-UVB given before and after transplantation of the melanocytes gives the best chance of repigmentation in active vitiligo patients. Interestingly, Yao et al. demonstrated more than 90% repigmentation at the 1-year follow-up in all patients treated with low-density cultured autologous melanocytes combined with NB-UVB treatment after cell transplantation [68]. Excellent repigmentation (85–100%) was also achieved in the small four-patient study where the non-cultured autologous melanocytes and keratinocytes transplantation was combined with UVA or UVB therapy after grafting [69].

Platelet-rich plasma (PRP) originates from the collection of venous blood, which is then centrifuged in the presence of anticoagulants. After centrifugation, autologous platelets are suspended in a small amount of plasma. Topical application using an intradermal injection of PRP through the secretion of platelet’s alpha granules increases the release of growth factors (especially basic fibroblast growth factor, bFGF), adhesion molecules, and chemokines, which, by interacting with the local environment, stimulate melanocyte migration along with the stimulation of keratinocyte and fibroblast proliferation. Moreover, PRP promotes the release of inflammatory mediators and modulators through the release of numerous anti-inflammatory cytokines, such as interleukins (IL-4, IL-10, IL-13), IL-1 receptor antagonist (IL-1ra), soluble tumor necrosis factor (TNF) receptor (sTNF-R) I, and interferon-gamma (IFN-γ). Although intralesional injection of PRP alone did not induce repigmentation, a combination of PRP with NB-UVB induced statistically significant repigmentation in a series of 60 patients from Egypt [70]. An interesting and promising study was performed with NCES suspended in PRP. Parambath et al. [71] compared the extent of repigmentation achieved using the transplantation of NCES in PRP and NCES in phosphate-buffered saline (PBS) in 21 patients with stable vitiligo. After 6 months, the repigmentation was 75.6% after NCES in PRP and 65% after NCES and PBS treatment (*p* = 0.0036). Moreover, the suspension in PRP was better assessed by patients in the visual analog scale. The clinical applications of combination therapy for vitiligo are summarized in Table 1 and Figure 2.

## 7. Are Cell-Based Therapies Appropriate for All Vitiligo Patients? Limitations and Challenges

Although vitiligo cell-based therapy is safe, well-tolerated, and effective at repigmentation with matching color and texture in appropriate candidates, it is still an underperformed treatment. There are several reasons for this: a limited number of practitioners know the technique details, a lack of awareness on the part of physicians, and a lack of insurance coverage for vitiligo because many consider it a cosmetic disease. Moreover, not all patients are willing to undergo this type of therapy. Parambath et al. [71] found that 11 out of 38 patients screened for the study were not willing to undergo surgery. When asked about the reasons for refusal, patients indicated the desire to receive a trial of medical therapy from a tertiary care center, fear of surgery, unwillingness for follow-up visits, and high costs. It should also be noted that not every vitiligo patient is eligible for surgical therapy. Depending on the clinical practice, most patients must have clinically stable vitiligo for 6 months to one year to qualify [45,58]. Clinically stable vitiligo is determined by the non-appearance of new lesions and by the absence of changes in the existing ones. Patients with segmental or focal vitiligo are better candidates because they tend to achieve greater repigmentation than those with generalized disease. Higher rates of repigmentation were also reported in young patients compared to older ones [72].

It is also very important to note that it is difficult to estimate the overall percentage of people who have recovered from vitiligo, as only a fraction of them undergo treatment. Moreover, even though combination therapy is more effective than monotherapy, recurrence affects up to 40% of patients [73].

Patients are excluded from surgery if they had a history of koebnerization, hypertrophic scarring, keloids, or are susceptible to poor wound healing. According to Ramos et al., isolated scalp leukotrichia is an adverse prognostic sign, and the presence of significant distal fingertip, periorificial, or acrofacial involvement is also an exclusion criterion because these disease variants typically respond poorly to the melanocyte–keratinocyte transplantation procedure [45]. However, as we discuss below, patients with leucotrichia in vitiliginous patches may also have a chance for successful treatment [59]. Recently, molecular markers were suggested for better predictions of vitiligo diagnosis and response to treatment. The RNA sequencing in vitiligo patients showed differences in expression levels of 470 genes between the skin specimens of responder versus non-responder patients. Two hundred sixty-nine upregulated genes were involved in processes, such as fatty acid omega oxidation, whereas down-regulated genes (two hundred and one) were related to PPAR and estrogen signaling pathways [74].

## 8. Conclusions

In conclusion, cell therapies undergo continuous improvements both toward better re-pigmentation effects and simplifying the methods, making them more accessible to dermatological clinics [75]. Modifications of procedures involving simplifying cell collection, ensuring their good transplantation potential, as well as using less sophisticated laboratory equipment, reduce the cost of the procedure and make it more accessible to patients. Another important factor when considering cell-based vitiligo treatment is the selection of appropriate candidates. In the meta-regression analyses by Ju et al. [75], the successful outcome (>90% repigmentation) was associated with younger age, segmental vitiligo, and a non-acral area.

It should also be mentioned that repigmentation after cell therapy progresses gradually and may continue beyond 12 months following the procedure. Thus, there is a great need for more extended follow-up studies (minimum 6 months) for evaluation of the effectiveness and real cost that the patients must undertake on their way along the treatment of vitiligo.

**Table 1 ijms-24-03357-t001:** The clinical applications of melanocyte transplantation, melanocyte–keratinocyte cell transplantation, ReCell, non-cultured epidermal cell grafting, and combination therapy for vitiligo.

Therapy	Vitiligo Characteristics	Patients Number	Age Range (Years)	Clinical Results	Evaluation Time	Reference	Year
Melanocyte transplantation							
Cultured autologous melanocytes	Stable vitiligo	25	13–72	Almost complete repigmentation in 6 out of 11 cases, in 4 cases, 40–71% of the grafted achromatic area was repigmented	1–20 months	Chen et al. [16]	2000
Autologous transplanted epidermal cell suspensions	Stable vitiligo or vitiligo with doubts about stability	28	5–65	A total of 77% repigmentation	12 months	Van Geel et al. [1]	2004
Autologous cultured pure melanocyte suspension	Stable vitiligo	120	7–72	Overall, 84% of patients with localized vitiligo experienced 90% to 100% coverage	6–66 months	Chen et al. [28]	2004
Autologous melanocyte rich cell suspension (non-cultured) and cultured melanocyte technique	Stable vitiligo	27	21–30	Over 90% repigmentation in 52.17% of cases with the autologous melanocyte-rich cell suspension technique and in 50% with the melanocyte culture technique	Up to 6 months	Pandya et al. [12]	2005
Autologous cultured pure melanocytes	Stable vitiligo for at least 6 months	102	8–12, 13–17, Adults: mean 29	Repigmentation of 50% or more in children, adolescents, and adults were 83.3%, 95.0% and 84.0% cases, respectively	At least 6 months	Hong et al. [17]	2011
Autologous melanocytes culture on a denuded amniotic membrane	Stable vitiligo	4	13–29	A total of 90–95% repigmentation in all patients	Up to 6 months	Redondo et al. [18]	2011
Autologous melanocytes transplantation	Stable vitiligo and developing vitiligo	16	19–40	In 87.5% of lesions, a repigmentation of >50%; no relapse was observed	5 years	Zhu et al. [26]	2017
Autologous epidermal cell suspension	Stable vitiligo	300	Range not stated, mean 27.1	Repigmentation stability remained in most treated patches	Up to 30 months	Orouji et al. [35]	2017
Autologous non-cultured and trypsinized melanocyte grafting	Stable vitiligo	28	Range not stated, mean 25.9	Over 50% in the face and neck, trunk, upper extremity, and genitals in 57.4%, 20.4%, 16.7%, and 5.5% patients, respectively	18 months	Ghorbani et al. [14]	2022
Hair follicle cell transplantation							
Non-cultured extracted hair follicular outer root sheath (ORS) cell suspension transplantation	Stable vitiligo for at least 3 months	14	17–32	Nine out of fourteen patients achieved >75% repigmentation	1–15 months	Mohanty et al. [19]	2011
Autologous non-cultured outer root sheath hair follicle cell suspension (NCORSHFS)	Stable vitiligo	30	8–38	The number of melanocytes and HFSC transplanted were significantly higher among patients achieving optimum (>75%) repigmentation	24 weeks	Vinay et al. [20]	2015
Hair follicle outer root sheath cell transplantation (EHF ORS)	Stable vitiligo	20	18–43	Mean repigmentation of 80.15% with 90–100% in 60% of patients	6 months	Shah et al. [21]	2016
Non-cultured, extracted follicular outer root sheath suspension (NC-EHF-ORS-CS)	Stable vitiligo	2	18–36	The mean repigmentation was 52% and >75% repigmentation in 32% of patients	6 months	Kumar et al. [22]	2018
Autologous hair follicle cell derived melanocytes transplantation	Stable vitiligo	26	19–50	Overall, 34.6% of patients achieved excellent repigmentation, 50% had good, 11.5% had fair, and 3.9% had poor repigmentation	1 year	Shi et al. [23]	2020
Keratinocyte transplantation							
Autologous cultured keratinocytes	Stable vitiligo	27	9–48	Twelve patients had 90% or more repigmentation after the first surgery, which increased by two cases when patients with multiple surgeries were included	At least 1 year	Matsuzaki and Kumagai [30]	2013
Melanocyte–keratinocyte transplantation							
Cultured epithelial autographs with keratinocytes seeded at high density	Stable or active vitiligo	5	32–71	Out of five patients, repigmentation was only achieved in one	N/A	Phillips et al. [33]	2001
Melanocyte–keratinocyte cell transplantation (MKT)	Stable vitiligo	184	9 to 70	Overall, 53% in the generalized vitiligo and 84% in the segmental vitiligo group showed 95–100% repigmentation	Up to 1 year	Mulekar et al. [43]	2003
Autologous non-cultured melanocyte–keratinocyte cell transplantation (MKT)	Stale vitiligo for at least 1 year	134	At least 12	Overall, 84% in the segmental and 73% in the focal vitiligo group showed 95–100% repigmentation	Up to 5 years	Mulekar et al. [47]	2004
Autologous non-cultured melanocyte–keratinocyte cell transplantation (MKT)	Stable genital vitiligo	3	24–39	Near-complete repigmentation observed in all patients	Up to 1 year	Mulekar et al. [49]	2005
Autologus melanocyte–keratinocyte cell transplantation (MKT)	Stable vitiligo (for at least 6 months)	142	18–70	Overall, 56% patients showed 95–100% repigmentation	Up to 6 years	Mulekar et al. [48]	2005
Noncultured melanocyte–keratinocyte cell transplantation (MKT)	Stable vitiligo (for at least 6 months)	49	7–65	For bilateral vitiligo, more than 50% of patients showed >65% repigmentation. For unilateral vitiligo, all but two patients treated for the eyelids vitiligo showed >65% repigmentation	N/A	Mulekar et al. [55]	2009
Autologous dissociated epidermal cell suspensions	Stable vitiligo	10	17–52	Overall, 76–100% repigmentation in 40% of patients	6 months	Khodadadi et al. [34]	2010
Non-cultured melanocyte–keratinocyte transplantation (MKT)	Stable vitiligo	25	8–45	Overall, 23% of patients showed 90–100% repigmentation	6–17 months	El-Zawahry et al. [56]	2011
Non-cultured melanocyte–keratinocyte transplantation (MKT)	Stable vitiligo	8	13–43	Of eight lesions treated with non-cultured MKT, four lesions showed 96–100%, one lesion 65–95%, and three lesions 0–25% repigmentation	4 months	Toossi et al. [50]	2011
Melanocytes and keratinocytes transplantation using the sandpaper technique combined with dermabrasion or only dermabrasion	Stable vitiligo	11	21–65	At the end of treatment, both techniques showed similar repigmentation with repigmentation in nine cases (87% to 6%) of pigmenation in the transfer group and nine cases (94% to 5%) in the dermabrasion group	3 months	Quezada et al. [52]	2011
Melanocyte–keratinocyte cell suspension with dermabrasion (MKT+ DA) and dermabrasion alone (DA)	Stable vitiligo	11	35–48	Slightly better pigmentation occurred with DA+MKT in 7 out of 11 patients	12 months	Vazquez-Martinez et al. [51]	2011
Autologous melanocyte–keratinocyte transplantation (MKT)	Stable vitiligo	23	18–60	Overall, 95–100% repigmentation in 17%	3–6 months	Huggins et al. [53]	2012
Autologous transplantation of non-cultured melanocyte–keratinocyte cell suspension (MKT)	Stable vitiligo	20	10–50	Overall, 25% of patients showed ≥90% Repigmentation; the best results were observed in face and neck	Up to 24 months	Ramos et al. [58]	2017
Autologous melanocyte–keratinocyte transplantation (MKT)	Stable vitiligo without fingertip involvement	100	9–60	MKT could maintain repigmentation for at least 72 months	12–72 months	Silpa-Archa et al. [44]	2017
Autologous non-cultured keratinocyte–melanocyte suspension (MKT)	Stable vitiligo	5	Range not stated, mean 20	Overall, 76–100% repigmentation in 60% of patients	6 months	Benzekri and Gauthier [25]	2017
Autologous non-cultured melanocyte–keratinocyte transplantation (MKT)	Stable vitiligo	602	4–67	Overall, 84.3% of patients achieved ≥50% repigmentation at the 6th month evaluation; at 6 years, 23% showed relapse	6 years	Altalhab et al. [46]	2019
Combination therapies							
Non-cultured autologous melanocytes and keratinocytes combined with UVA or UVB	Stable vitiligo	4	30–52	Overall, 85–100% repigmentation achieved at 6 to 20 months	6–20 months	van Geel et al. [68]	2001
Narrow-band ultraviolet B therapy for cultured autologous melanocyte transplantation patients	Stable vitiligo	437	5–55	A total of 20 sessions of NB-UVB treatment before transplantation and 30 sessions after transplantation gave the best repigmentation	6 months	Zhang et al. [66]	2014
Cultured autologous melanocyte transplantation (CMT combined with narrowband ultraviolet B (NB-UVB)	Stable vitiligo	8	7–28	All patients treated with low-density CMT combined with NB-UVB obtained more than 90% repigmentation	1 year	Yao et al. [67]	2017
Platelet-rich plasma (PRP) used to suspend non-cultured epidermal cell suspension (NCES) before transplantation	Stable vitiligo	21	Range not stated, mean 23.1	Suspending NCES in PRP can result in significantly greater mean repigmentation	6 months	Parambath et al. [70]	2019
ReCell							
ReCell vs. conventional melanocyte–keratinocyte transplantation (MKT)	Stable vitiligo	5	18–40	Overall, 40% of lesions treated with ReCell showed 100% repigmentation, while 20% of lesions failed to repigment. Overall, 60% of lesions treated with conventional MKT showed 100% repigmentation and 20% failed to repigment	4 months	Mulekar et al. [62]	2008
ReCell	Stable vitiligo	15	18–45	Repigmentation greater than 75% was recorded in 12 (80%) patients	Minimum 6 months	Cervelli et al. [61]	2009
ReCell	Stable vitiligo	1	N/A	The patient had >90% repigmentation	N/A	Cervelli et al. [62]	2010
Non-cultured cellular grafting							
Autologous, non-cultured cellular grafting (MKTP)	Stable vitiligo for at least 6 months	25	4–16	Overall, 95–100% repigmentation in 62% of patients	9–54 months	Mulekar et al. [60]	2010
Autologous non-cultured epidermal suspension (NCES)	Stable vitiligo	13	8–17	Overall, 79% of lesions had >90% repigmentation	1 year	Sahni et al. [61]	2011
Non-cultured cellular grafting	Stable vitiligo	13	15–52	Overall, 91% of the patients achieved >50% repigmentation	3–12 months	Gan et al. [59]	2012
Non-cultured epidermal cell suspension (NCES)	Stable vitiligo	36	16–47	More than 75% repigmentation in 63.75% of lesions	6–18 months	Holla et al. [57]	2013
Non-cultured epidermal cell suspension (NCES)	Stable vitiligo	37	Range not stated, mean: 28.3 (group 1) 24.1(group 2) 22.4 (group 3)	Cell count significantly lower in the ORSHFS compared with NCES with no significant difference in the repigmentation outcome	18 months	El-Zawahry et al. [39]	2017
Autologous non-cultured epidermal cell suspension (NCES)	Stable vitiligo	41	8–50	Overall, 80.5% of patients showed 51–75% repigmentation, and 17.1% showed complete or almost complete repigmentation	6–9 months	Liu et al. [72]	2019
Non-cultured autologous epidermal cell grafting resuspended in hyaluronic acid (A ready-to-use kit, Viticell^®^)	Stable vitiligo	36	17–67	For difficult-to-treat lesions, no repigmentation ≥50% was observed; for other locations, the success rate was significantly higher	12 months	Bertolotti et al. [76]	2020
Comparative studies							
Three different transplantation methods: autologous cultured melanocytes, ultrathin epidermal sheets, and basal layer cell suspension	Stable or unstable vitiligo	132	8–61	Stable vitiligo patients responded in most cases with 100% repigmentation in all studied treatments	1–7 years	Olsson et al. [27]	2002
Autologous non-cultured epidermal cell suspension (NCES) compared to suction blister epidermal grafting (SBEG)	Stable vitiligo	41	12–40	Overall, 90–100% repigmentation in 71% of lesions in the NCES group and 27% of lesions in the SBEG group	4 months	Budania et al. [37]	2012
Autologous non-cultured epidermal cell suspension (NCES) compared to autologous non-cultured extracted hair follicle outer root sheath cell suspension (NCORSHFS)	Stable vitiligo	30	13–35	Overall, 90–100% repigmentation in 83% of lesions in the NCES group and 65% of lesions in the NCORSHFS group	4 months	Singh et al. [38]	2013
Comparison between autologous melanocyte rich cell suspension (NCMT technique) and cultured melanocyte technique (CMT)	Stable vitiligo	25	N/A	More than 70% repigmentation in 62.17% cases treated with NCMT and in 52% with the CMT	6 months	Verma et al. [13]	2014
Blister roof grafting (BG), cultured melanocytes transplantation (CMT) and non-cultured epidermal cell suspension transplantation (NCES)	Stable vitiligo	83	Range not stated, mean 25	More than 50% repigmentation in 92%, 82%, and 81% of the 83 patients treated with the BG, CMT, and NCES methods, respectively	12 months	Bao et al. [36]	2015
Autologous cultured melanocytes transplantation (CMT) and non-cultured epidermal cell suspension transplantation (NCES)	Stable vitiligo	30	Range not stated, mean 26.1	Overall, 66.7% of cases showed more than 70% repigmentation with CMT; NCES resulted in less than 40% repigmentation in most of the cases	3–6 months	Verma et al. [40]	2015
Excimer laser alone compared to non-cultured melanocyte–keratinocyte transplantation (MKCT) alone and combination therapy	Stable vitiligo	10	21–48	Excimer laser combined with non-cultured MKCT improves the repigmentation rate, with an average of 41.9% reduction of depigmented area surface	2 weeks	Ebadi et al. [54]	2015
Epidermal melanocyte transfer (EMT) compared to hair follicular melanocyte transfer (HFMT)	Stable vitiligo	11	12–42	More than 90% repigmentation observed in 83.33% patches of the EMT group and 43.33% in the HFMT group	6 months	Donaparthi et al. [77]	2016
Autologous non-cultured epidermal cell suspension (ECS) and follicular cell suspension (FCS)	Stable vitiligo	5	21–33	Superior repigmentation obtained in combined ECS and FCS treatment	4 months	Razmi et al. [41]	2017
Hair follicle transplantation (follicular unit transplantation, FUT) and autologous non-cultured, non-trypsinized epidermal cells grafting (Jodhpur Technique-JT)	Stable vitiligo	30	21–25	More than 75% repigmentation was observed in 70% lesions in the FUT group and 72% of lesions in the JT group	20 weeks	Lamoria et al. [24]	2020

## Figures and Tables

**Figure 1 ijms-24-03357-f001:**
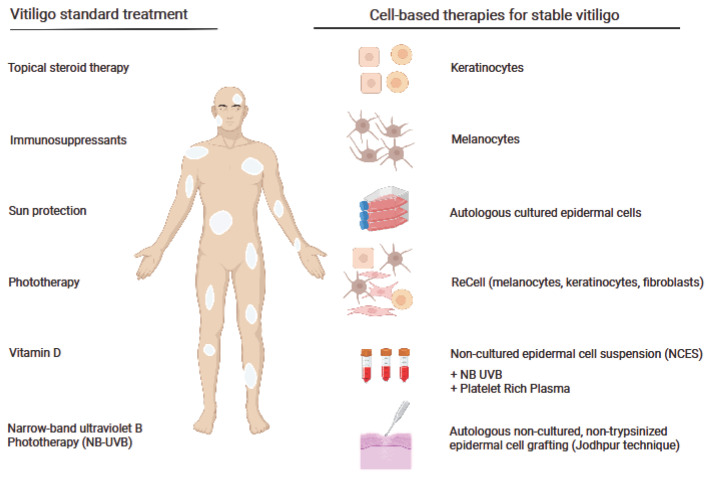
Standard and cell-based therapies for vitiligo. The figure was created with Biorender.com, accessed on 3 November 2022.

**Figure 2 ijms-24-03357-f002:**
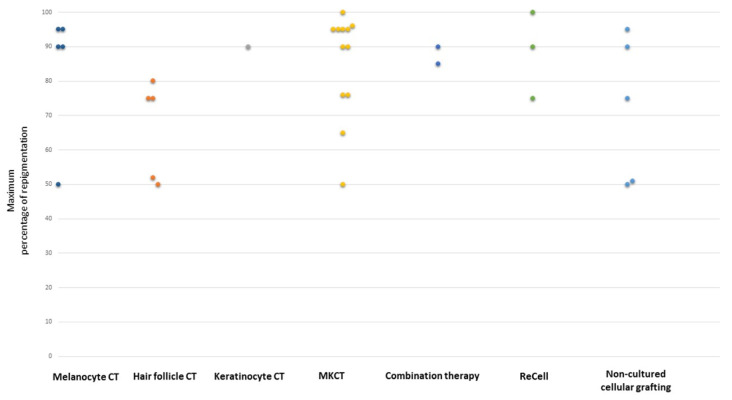
CT-cell therapy. The highest percentage of repigmentation depending on the cell-based therapy used (based on papers presented in Table 1).

## Data Availability

No new data were created or analyzed in this study. Data sharing is not applicable to this article.

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
