# Peer review of "Current Status of Cell-Based Therapies for Vitiligo"

_ijms, 2023, doi:10.3390/ijms24043357_

Round 1

Reviewer 1 Report

The overall impression of the technical contribution of the current study is marginal. However, the Authors may consider making necessary amendments to the manuscript for better comprehensibility of the study.

1. Abstract fails to give clear insight on the contributions of the current study, abstract must be re-written, focusing on the technical aspects of the proposed model, the main experimental/survey results, and the metrics used in the evaluation of the current status.

2.  The contribution of the current study must be briefly discussed as bullet points in the introduction. And motivation must also be discussed in the manuscript.

3. The overall organization of the manuscript is not discussed anywhere in the manuscript. Please add the same in the introduction section of the manuscript.

4. All web references like (https://pubmed.ncbi.nlm.nih.gov/) are to be added in references with the last date of access and must be cited in the text.

5. From the statement "relatively low 68 (57.4%)" at line 69, why are the values in enclosed within the parenthesis.? Does it have any special meaning?

6. why some of the studies are taken way back from 2004, they must be obsolete and many new technologies/techniques have come in recent times. 

7. Authors are recommended to provide citations for the claims like "However, results have been inconsistent in some studies, a" at line 147. 

8. Any graphs that show the impact of various therapy for vitiligo for ease of understanding of statistical analysis. The manuscript is full of text-based explanations with lesser statistical analysis. 

9. what is the summary of the study and where is it concluded?

10. what are the practical implications involved in the current study? where are they discussed?

11. overall what percentage of affected people have recovered from vitiligo.? Any census regarding the same.

12. Does the study provide adequate information for the reader to re-produce the work? what are the insights does someone get from the current manuscript? Harly I could find anything significant.  sections 4, 5, 6, and 7 are available across the studies. 

Reviewer 2 Report

Updated review of the treatment of vitiligo and the role of cell cultures. It may well be the future of treatment for diseases such as vitiligo. The authors have done a thorough job, I have no comments to add.

Reviewer 3 Report

Normal pigmentation pattern is a critical component of the appearance of a person. Vitiligo is the most frequently occurring depigmentation disorder skin disease, and it can be developed at any age or gender. The authors first summarized the latest 20 years of cell-based therapy in human patients from paper and clinical trials. However, there are 2 minor concerns that need to be answered.

Is the treatment difference between non-segmental vitiligo and segmental vitiligo in clinic?

Can author discuss more limitations of the listed cell-based therapies?

Round 2

Reviewer 1 Report

Authors must highlight the changes done in the revised manuscript. The manuscript in its current form is challenging to track the amendments done by the authors. 

Highlight with different colors for each of the reviewer's comments, for ease of assessment. 

Author Response

Please see the attached manuscript with tracked changes.

Round 3

Reviewer 1 Report

The authors have done the necessary amendments as recommended by the reviewer. Manuscript in the current form may be considered for the next phase of the editorial process.